# Symmetry-Based Disentangled Representation Learning requires Interaction with Environments

Hugo Caselles-Dupré[1,2], Michael Garcia-Ortiz[2], David Filliat[1]

[1]Flowers Laboratory (ENSTA Paris & INRIA), [2]AI Lab (Softbank Robotics Europe)
caselles@ensta.fr, mgarciaortiz@softbankrobotics.com, david.filliat@ensta.fr

## Abstract

Finding a generally accepted formal definition of a disentangled representation in the context of an agent behaving in an environment is an important challenge towards the construction of data-efficient autonomous agents. Higgins et al. (2018) recently proposed Symmetry-Based Disentangled Representation Learning, a definition based on a characterization of symmetries in the environment using group theory. We build on their work and make observations, theoretical and empirical, that lead us to argue that Symmetry-Based Disentangled Representation Learning cannot only be based on static observations: agents should interact with the environment to discover its symmetries. Our experiments can be reproduced in Colab [1] and the code is available on GitHub [2].

## 1 Introduction

Disentangled Representation Learning aims at finding a low-dimensional vector representation of the world for which the underlying structure of the world is separated into disjoint parts (i.e., disentangled) reflecting the compositional nature of the said world. Previous work (Higgins et al., 2017; Raffin et al., 2019) has shown that agents capable of learning disentangled representations can perform data-efficient policy learning. However, there is no generally accepted formal definition of disentanglement in Representation Learning, which prevents significant progress in this emerging field.

Recent efforts have been made towards finding a proper definition (Locatello et al., 2019b). In particular, Higgins et al. (2018) defines Symmetry-Based Disentangled Representation Learning (SBDRL), by taking inspiration from the successful study of symmetry transformations in Physics. Their definition focuses on the transformation properties of the world. They argue that transformations that change only some properties of the underlying world state, while leaving all other properties invariant, are what gives exploitable structure to any kind of data. They distinguish between linear and non-linear disentangled representations, which models whether the transformation affects the representation in a linear or non-linear way. Supposedly, linearity should be more useful for downstream tasks such as Reinforcement Learning or auxiliary prediction tasks. Their definition is intuitive and provides principled resolutions to several points of contention regarding what disentanglement is. For clarity, we refer to a representation as SB-disentangled if it is disentangled in the sense of SBDRL, and as LSB-disentangled if linear disentangled.

We build on the work of Higgins et al. (2018) and make observations, theoretical and empirical, that lead us to argue that SBDRL requires interaction with environments. The necessity of having interaction has been suggested before (Thomas et al., 2017). We propose a proof for SBDRL.

As in the original work, we base our experiments on a simple environment, where we can formally define and manipulate a SB-disentangled representation. This simple environment is 2D, composed of one circular agent on a plane that can move left-right and up-down. The world is cyclic: whenever the agent steps beyond the boundary of the world, it is placed at the opposite end (e.g. stepping up at the top of the grid places the object at the bottom of the grid).

We prove, for this environment, that the minimal number of dimensions of the representation required for it to be LSB-disentangled is counter-intuitive (i.e. 4). Indeed, the natural number of dimensions required to describe the state of the world (i.e. 2) is not enough to describe its symmetries in a linear way, which is supposedly ideal for subsequent tasks. Additionally, learning a non-linear SB-disentangled representation is possible, but current approaches are not designed to model the effect of the world's symmetries on the representation, a key aspect of SBDRL which we present later. We thus ask: how is one supposed to, in practice, learn a (L)SB-disentangled representation?

We propose two approaches that arise naturally, one where representation and world symmetries effect on it are learned separately and one where they are learned jointly. For both scenarios, we formally define what could be a proper representation to learn, using the formalism of SBDRL. We propose empirical implementations that are able to successfully approximate these analytically defined representations. Both empirical approaches make use of transitions $(o_t, a_t, o_{t+1})$ rather than still observations $o_t$, which validates the main point of this paper: Symmetry-Based Disentangled Representation Learning requires interaction with the environment.

Ultimately, the goal of such representations is to facilitate the learning of downstream tasks. We study the efficiency of (L)SB-disentangled representation on a particular downstream task: learning an inverse model. Our results suggests that (L)SB-disentangled indeed facilitates the learning of such downstream task.

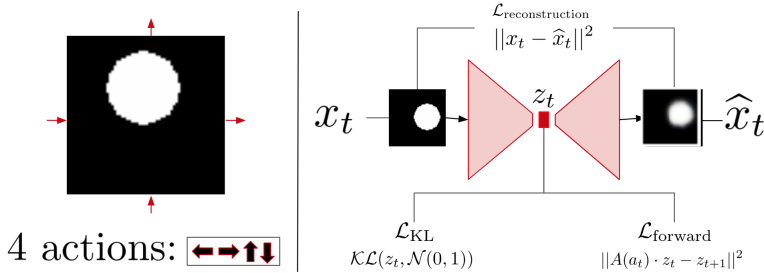

Figure 1: **Left:** Environment studied in this paper. **Right:** Proposed architecture for learning a LSB-disentangled representation in the environment at the left as presented in section 6.2.

Our contributions are therefore the following:

- We prove that interaction with the environment, i.e. the use of transitions, is necessary for SBDRL, and illustrate it empirically.

- We propose two alternatives for learning linear and non-linear SB-disentangled representation in practice, both using transitions rather than still observations. Using a simple environment, we describe both solutions theoretically and validate them empirically.

- We empirically demonstrate the efficiency of using SB-disentangled for a downstream task (learning an inverse model).

## 2 Symmetry-Based Disentangled Representation Learning

Higgins et al. (2018) defines Symmetry-Based Disentangled Representation Learning (SBDRL) in an attempt to formalize disentanglement in Representation Learning. The core idea is that SB-disentanglement of a representation is defined with respect to a particular decomposition of the symmetries of the environment. Symmetries are transformations of the environment that leave some aspects of it unchanged. For instance, for an agent on a plane, translations of the agent on the $y$-axis leave its $x$ coordinate unchanged. They formalize this using group theory. Groups are composed of

these transformations, and group actions are the effect of the transformations on the state of the world and representation.

The proposed definition of SB-disentanglement supposes that these symmetries are formally defined as a group $G$ (equipped with composition) that can be decomposed into a direct product $G = G_1 \times .. \times G_n$. We now recall the formal definition of a SB-disentangled representation w.r.t to this group decomposition. We advise the reader to refer to the detailed work of Higgins et al. (2018) for any clarification. Let $W$ be a set of world-states $W = (w_1, .., w_m) \in \mathbb{R}^{m \times d}$, where each state $w_i$ is a d-dimensional vector. We suppose that there is a generative process $b : W \to O$ leading from world-states to observations (these could be pixel, retinal, or any other potentially multi-sensory observations), and an inference process $h : O \to Z$ leading from observations to an agent's representations. We consider the composition $f : W \to Z, f = h \circ b$. Suppose also that there is a group $G$ of symmetries acting on $W$ via a group action $\cdot_\mathcal{W} : G \times W \to W$. A world is thus defined by $(W, \cdot_\mathcal{W})$. We would like to find a corresponding group action $\cdot_Z : G \times Z \to Z$ so that the symmetry structure of $W$ is reflected in $Z$. We also want the group action $\cdot_Z$ to be disentangled, which means that applying $G_i$ to $Z$ leaves all sub-spaces of $Z$ unchanged but the one corresponding to the transformation $G_i$. Formally, the representation $Z$ is SB-disentangled with respect to the decomposition $G = G_1 \times .. \times G_n$ if:

1. There is a group action $\cdot_Z : G \times Z \to Z$.
2. The map $f : W \to Z$ is equivariant between the group actions on $W$ and $Z$:

$$\boxed{g \cdot_Z f(w) = f(g \cdot_\mathcal{W} w)} \qquad \leftrightarrow \qquad \begin{array}{ccc} G \times W & \xrightarrow{\cdot_W} & W \\ {\scriptstyle id_G \times f} \downarrow & & \downarrow {\scriptstyle f} \\ G \times Z & \xrightarrow{\cdot_Z} & Z \end{array}$$

3. There is a decomposition $Z = Z_1 \times .. \times Z_n$ such that each $Z_i$ is fixed by the action of all $G_j, j \neq i$ and affected only by $G_i$.

This definition of SB-disentangled representations does not make any assumptions on what form the group action should take when acting on the relevant disentangled vector subspace. However, many subsequent tasks may benefit from a SB-disentangled representation where the group actions transform their corresponding disentangled subspace linearly. Such representations are termed linear SB-disentangled representations, which we refer to as LSB-disentangled representations.

## 3 Symmetry-Based Disentangled Representation Learning requires interaction with environments

In this section we prove the main claim of this paper: SBDRL requires interaction with environments. By "interaction with environments" we refer to the fact that in order to learn a SB-disentangled representation, one should **not** use a training set composed of still samples $(o_t, o_{t+1}, ...)$, but rather transitions $((o_t, a_t, o_{t+1}), (o_{t+1}, a_{t+1}, o_{t+2}), ...)$.

We begin by observing that SBDRL definition is actually two-fold. The definition of a SB-disentangled representation w.r.t the decomposition $G = G_1 \times .. \times G_n$ is composed of two main properties:

1. There is a group action $\cdot_Z : G \times Z \to Z$.
2. The map $f : W \to Z$ is equivariant between the group actions on $W$ and $Z$.

$\left.\right\}$ Definition of a Symmetry-Based representation.

3. There is a decomposition $Z = Z_1 \times .. \times Z_n$ such that each $Z_i$ is fixed by the action of all $G_j, j \neq i$ and affected only by $G_i$.

$\left.\right\}$ Disentanglement property.

The first two points define what a SB representation is. It's a representation for which the effect of group actions on the world state is the same as the effect on the representation itself. The third point characterizes what disentanglement is for a SB representation.

In practice, it seems natural to first know how to learn a representation that satisfy the first two points, i.e. a SB representation. Based on this, we can develop methods that enforce disentanglement.

Hence we ask, how can one learn a SB representation? This task involves knowledge about how the group action affect $Z$. The group action is defined to be the effect of symmetries on the representation. These symmetries can be translations, rotations, time translations, etc. In a Machine Learning paradigm, we would design an algorithm that learns from examples. We thus need, in practice, a way to apply these transformations on observations of the world $(o_t)_{t=1..n}$ and observe the result $(g_t \cdot_Z o_t = o_{t+1})_{t=1..n}$.

We thus make an analogy between the effect of a symmetry $g$ (by the group action $\cdot_{\mathcal{W}}$) on the environment $(o_1, g, g \cdot_{\mathcal{W}} o_1 = o_2)$, and a transition that uses the dynamics $f$ of the environment $(o_t, a_t, f(o_t, a_t) = o_{t+1})$. It allows us to consider a more realistic scenario where we have an agent in an environment, and we can apply the group actions to this agent. In our analogy we simply say that $o_1 = o_t$, $o_2 = o_{t+1}$ and $a_t = g$ and $\cdot_{\mathcal{W}} = f$.

However, we do not make a total confusion between symmetries and regular actions that can be found in any environment. A symmetry is an element of a group (in the mathematical sense) of functions $g : W \to W$, and the binary operation of the group is composition. In that sense, these functions can effectively be considered as actions, because actions take the environment from one state to another through the dynamics $f$, and symmetries take the environment from one state to another through the group action $\cdot_{\mathcal{W}}$.

It is important to mention that not all actions are symmetries, for instance the action of eating a collectible item in the environment is not part of any group of symmetries of the environment because it might be irreversible.

More formally, Theorem 1 provides a mathematical proof that we need interaction with environments.

**Theorem 1.** *Suppose we have a SB representation $(f, \cdot_Z)$ of a world $\mathcal{W}_0 = (W = (w_1, .., w_m) \in \mathbb{R}^{m \times d}, \cdot_{\mathcal{W}_0})$ w.r.t to $G = G_1 \times ... \times G_n$ using a training set $\mathcal{T}$ of unordered observations of $\mathcal{W}_0$. Let $W_k$ be the set of possible values for the $k^{th}$ dimension of $w \in W$.*
*Then:*

1. *There exists at least $k_{W,G} = n[(\min_k(card(W_k))!] - 1$ worlds $(\mathcal{W}_1, .., \mathcal{W}_{k_{W,G}})$ equipped with the same world states $\mathcal{W}_i = (w_1, .., w_m)$ and symmetries $G$, but different group actions $\cdot_{\mathcal{W}_i}$.*

2. *For these worlds, $(f, \cdot_Z)$ is not a SB representation.*

3. *These worlds can produce exactly the same training set $\mathcal{T}$ of still images.*

*Proof.* Consider two identical worlds $(\mathcal{W}_1, \mathcal{W}_2)$ equipped with the same symmetries $G$. Suppose that they are given two different group actions $\cdot_{1,Z}$ and $\cdot_{2,Z}$ i.e. the effect of $G$ on $\mathcal{W}_1$ is not the same as its effect on $\mathcal{W}_2$. Then, given a fixed observation $o$, it is impossible to tell if $o$ is an observation of $\mathcal{W}_1$ or $\mathcal{W}_2$. It is only possible to tell if we have access to transitions $(o_t, g_t, o_{t+1})_{t=1..n}$ and observe the result $(g_t \cdot_Z z_t = z_{t+1})_{t=1..n}$. See Appendix A.1 for full proof.

Using Theorem 1, we can deduce that for a given dataset of still images collected in a world, it is impossible to describe the action of symmetries on the world. The dataset could come from a number of different worlds where symmetries act differently. Hence the need for transitions. For example, in a world where the agent can change color along a hue axis, the succession of colors can be (red, green, blue, red, ...), or (red, blue, green, red, ...). Then the world states are identical, the symmetries also. Yet, the effect of the symmetries are not the same, i.e. $\cdot_{1,W} \neq \cdot_{2,W}$.

Still, it is not clear how to discover the symmetries $G$ of a world. Higgins et al. (2018) propose to use active perception or causal manipulations of the world to empirically determine them. Having this in mind, we note that high-level actions in an environment often correspond to symmetries, such as translations along cartesian axis, rotations, changes of color, changes related to time (no-op action). Actions could then be used as replacement to symmetries, and one could learn SB representations using traditional transitions $(o_t, a_t, o_{t+1})_{t=1..n}$ that are readily available in most environments. In the rest of the paper, we validate this approach empirically.

## 4 Considered environment

In this paper, we consider a simplification of the environment studied in (Higgins et al., 2018). This environment is 2D, composed of one circular agent on a plane that can move left-right and up-down, see Fig.1. Whenever the agent steps beyond the boundary of the world, it is placed at the opposite end (e.g. stepping up at the top of the grid places the object at the bottom of the grid). The world-states can be described in two dimensions: $(x, y)$ position of the agent. All of our experimental results are based on this environment. It is simple, yet presents the basis for a navigation environment in 2D. We chose this environment because we are able to define theoretically SB-disentangled representations, without making any approximation. We implement this simple environment using Flatland (Caselles-Dupré et al., 2018). The code is available in Colab[3] and Github[4]. All architecture and hyperparameters details are specified in Appendix B.

## 5 Theoretical analysis

We first provide a theoretical analysis of what can be learned in the considered environment, in the formalism of SBDRL. Learning a non-linear SB-disentangled representation of dimension 2 is possible. If $(x, y)$ is the position of the object, then learning these two coordinates as well as the cyclical effect of translations is enough to create a SB-disentangled representation of dimension 2.

However, it is not the case for LSB-disentangled representations. We provide a theorem that proves it is impossible to learn a LSB-disentangled representation of dimension 2 in the environment presented in Sec.4 (the result also applies to the environment considered in Higgins et al. (2018)). The key element of the proof is that the two actual dimensions of the environment are not linear but cyclic. Hence the impossibility of modelling two cyclic dimensions using two linear dimensions. See Appendix A.2 for full proof of the result.

Based on this, we show next how to learn, in practice, a SB-disentangled representation of dimension 2 and a LSB-disentangled representation of dimension 4.

## 6 Symmetry-Based Disentangled Representation Learning in practice

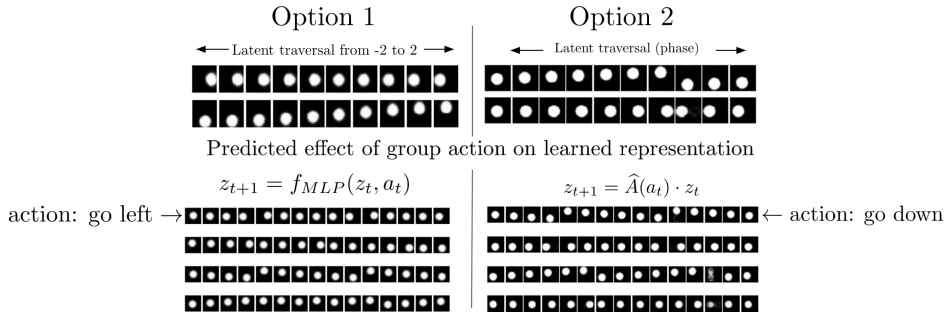

Figure 2: **Left:** First option: decoupled learning of representation and group action, here applied to learning a non-linear SB-disentangled representation. Latent traversal spanning from -2 to 2 over each of the representation's dimensions, followed by the predicted effect of the group action associated with each action (left, right, down, up). **Right:** Second option: joint learning of representation and group action, here applied to learning a non-linear LSB-disentangled representation. The representation is complex: latent traversal over the phase of each of the representation's dimensions, followed by the predicted linear effect of the group action associated each action (down, left, up, right).

We consider the problem of learning, in practice, SB-disentangled and LSB-disentangled representations for the world considered in Sec.4. For that, we propose two approaches: decoupled and end-to-end.

We illustrate each method by learning a SB-disentangled representation with the decoupled approach, and learning a LSB-disentangled representation with the end-to-end approach.

## 6.1 Decoupled approach (illustrated on SB-disentangled representation)

We propose to learn the representation first, and then the group action of $G$ on $Z$ using a separate model. This way, we have a complete description of the SB-disentangled representation. This approach is effectively decoupling the learning of physics from vision as in (Ha and Schmidhuber, 2018).

We consider learning a 2-dimensional SB-disentangled representation. We started by reproducing the results in (Higgins et al., 2018): we used a variant of current state-of-the-art disentangled representation learning model CCI-VAE. The learned representation corresponds (up to a scaling factor) to the world-state $W$, i.e. the $(x, y)$ position of the agent. This intuitively seems like a reasonable approximation to a disentangled representation.

However, once the representation is learned, we have no idea how the group action of symmetries affect the representation, even though it is at the core of the definition of SBDRL. This is where the necessity for transitions $(o_t, a_t, o_{t+1})_{t=1..n}$ rather than still observations $(o_t)_{t=1..n}$ comes into play. We learn the group action on $Z \cdot_Z : G \times Z \to Z$, such that $f = h \circ b$ is an equivariant map between the actions on $W$ and $Z$.

In practice, we learn $h : O \to Z$ with a variant of CCI-VAE, and then use a multi-layer perceptron to learn the group action on Z. The results are presented in Fig.2, where we observe that the learned group action correctly approximates the cyclical movement of the agent. We thus have learned a properly SB-disentangled representation of the world, w.r.t to the group decomposition $G = G_x \times G_y$.

## 6.2 End-to-end approach (illustrated on LSB-disentangled representation)

In the decoupled approach, the learned representation is identical to a setting where we would have ignored the group action. Hence, a preferable approach would be to jointly learn the representation and the group action. We study such approach on the task of learning a LSB-disentangled representation.

To accomplish this, we start with a theoretically constructed LSB-disentangled representation. It is based on a example given in Higgins et al. (2018). The representation is defined as following, using 4 dimensions:

- $f : \mathbb{R}^2 \to \mathbb{C}^2$ is defined as $f(x, y) = (e^{2i\pi x/N}, e^{2i\pi y/N})$

- $\rho(g) : \mathbb{C}^2 \to \mathbb{C}^2$ is defined as $\begin{cases} \rho(g_x)(z_x, z_y) = & (e^{2i\pi n_x/N} z_x, z_y) \\ \rho(g_y)(z_x, z_y) = & (z_x, e^{2i\pi n_y/N} z_y) \end{cases}$

In this representation, the $(x, y)$ position is mapped to two complex numbers $(z_x, z_y)$. For each translation (on the x-axis or y-axis), the associated group action on $Z$ is a rotation on a complex plane associated to the specific axis. This representation linearly accounts for the cyclic symmetry present in the environment.

Using CCI-VAE with 4 dimensions fails to learn this representation: we verified experimentally that only 2 dimensions were actually used when learning (for encoding the $(x, y)$ position), and the two remaining were ignored.

In order to learn the LSB-disentangled representation, we generate a dataset of transitions, and use it to learn the 4-dimensional LSB-disentangled representation with a specific VAE architecture we term Forward-VAE. This architecture allows to jointly learn the representation and the group action on it. Here, we want the group action on $Z$ to be linear, so we enforce linearity in transitions in the representation space.

We begin by re-writing the complex-valued function $\rho(g) : \mathbb{C}^2 \to \mathbb{C}^2$ as a real-valued function:

$$\rho(g) : \begin{array}{l} \mathbb{R}^4 \to \mathbb{R}^4 \\ v \to \rho(g)(v) = A^*(g) \cdot v \end{array} \tag{1}$$

where $A^*(g)$ is a 4x4 block-diagonal matrix, composed of 2x2 rotation matrices. Let's consider the environment in Sec.4. The agent has 4 actions: go left, right, up or down. We associate each action with a corresponding matrix with trainable weights.

For instance, if $g = g_x \in G_x$ is a translation on the x-axis, the corresponding matrix is $A^*(g_x)$ and we associate actions go right/left with corresponding matrices $\hat{A}(a_t)$, where $\cdot$ are trainable parameters:

$$A^*(g_x) = \begin{bmatrix} \cos(n_x) & -\sin(n_x) & 0 & 0 \\ \sin(n_x) & \cos(n_x) & 0 & 0 \\ 0 & 0 & 1 & 0 \\ 0 & 0 & 0 & 1 \end{bmatrix} \text{ and } \hat{A}(g_x) = \begin{bmatrix} \cdot & \cdot & 0 & 0 \\ \cdot & \cdot & 0 & 0 \\ 0 & 0 & 1 & 0 \\ 0 & 0 & 0 & 1 \end{bmatrix}.$$

We would like the representation model that we learn to satisfy $\rho(g)(v_t) = \hat{A}(g) \cdot v_t = v_{t+1}$. We thus enforce the representation to satisfy it in our Forward-VAE architecture, as illustrated in Fig.1. The training procedure is presented in Algorithm 1 in Appendix C.2 For each image in a batch, we compute $f(o_t) = z_t$ and $f(o_{t+1}) = z_{t+1}$ using the encoder part of the VAE. Then we decode $z_t$ with the decoder and compute the reconstruction loss $\mathcal{L}_{reconstruction}$ and annealed KL divergence $\mathcal{L}_{KL}$ as in (Caselles-Dupré et al., 2019). Then we compute $\hat{A}(a_t) \cdot z_t$ and compute the forward loss, which is the MSE with $z_{t+1}$: $\mathcal{L}_{forward} = (\hat{A}(a_t) \cdot z_t - z_{t+1})^2$. We then backpropagate w.r.t to the full loss function of Forward-VAE:

$$\mathcal{L}_{Forward-VAE} = \mathcal{L}_{reconstruction} + \gamma_t \cdot \mathcal{L}_{KL} + \mathcal{L}_{forward} \tag{2}$$

The results are presented in Fig.2 and Appendix D. Forward-VAE correctly learns a representation where the two complex dimensions correspond to the position $(x, y)$ of the agent. Plus, we observe that the learned matrices $(\hat{A}_i)_{i=1..4}$ are very good approximation of the ideal matrices $(A_i^*)_{i=1..4}$ defined above, with $n_x \approx \frac{\pi}{3}$. The mean squared difference is very small (order of $10^{-4}$).

## 6.3 Remarks

Note that we could have applied this joint learning approach to learning non-linear SB-disentangled representation. However it is not possible to apply the decoupled approach to learning a LSB-disentangled representation.

We used inductive bias given by the theoretical construction of a LSB-disentangled representation theory to design the action matrices and its trainable weights. This construction is specific to this example. However, the idea of having an action matrix for each action is extendable. If each action is high-level and associated to a symmetry, then SBDRL can be performed. Still, it requires high level actions that represent these symmetries. One potential way to find these actions is through active search (Soatto, 2011), as suggested in (Higgins et al., 2018).

In our Forward-VAE architecture we indeed explicitly design the model such that the resulting representation is Linear SB-disentangled, because we enforce linearity, force the representation to be SB (see points 1 and 2 in the definition in Sec.3) and by design have two separate subspaces for each symmetry. A more general approach would have been not to have those two separated subspaces and learn the entire action matrices, and thus we won't have the guarantee that the representation will satisfy the disentangled property. We ran this experiment and obtained the expected result: the learned representation is Linear-SB but not disentangled. This means that the x and y coordinates are not properly disentangled w.r.t to the considered group decomposition (i.e. a latent traversal over each dimension would not result in only a movement of the agent along the x or y coordinate). However the learned actions matrices are able to describe how the symmetries affect the representation and in a linear way. Hence enforcing disentanglement is the only viable option we found for LSB-disentanglement with this architecture.

It is important to note that an instability in Forward-VAE training can be expected due to the different contributions of the loss: at each training steps the goal of the forward part of the loss is to have a latent space that is suited for predicting $z_{t+1}$ using $z_t$. The rest of the loss is the VAE, which tries to learn a latent space that allows reconstruction. Hence the balance between these two seemingly unrelated objectives might be a source of instability. However it worked in practice, without any re-weighting of the objectives, which was a surprise.

# 7  Using (L)SB-disentangled representations for downstream tasks

Is using (L)SB-disentangled representations beneficial for subsequent tasks? This remains to be demonstrated, as other work have already challenged the benefit of learning disentangled representations over non-disentangled ones (Locatello et al., 2019b). In this section we wish to answer the following question: **is it increasingly better to use non-disentangled/non-linear SB-disentangled/LSB-disentangled representation for downstream tasks?** We define better in terms of final performance, under different settings (restricted capacity classifiers/restricted amount of data).

For the choice of downstream task, we select the task of learning an inverse model, which consists in predicting the action $a_t$ from two consecutive states $(s_t, s_{t+1})$.

As a LSB-disentangled representation models the interaction with the environment linearly, it intuitively should be increasingly easier to learn an inverse model from: a non-disentangled representation, a non-linear SB-disentangled representation, and a LSB-disentangled representation.

## 7.1  Experimental protocol

In order to test this hypothesis, we selected a well-established implementation (Scikit-learn (Pedregosa et al., 2011)) of a well-studied classifier (Random Forest (Breiman, 2001)). We collect 10k transitions $(o_t, a_t, o_{t+1})$. We train the following models and baselines to compare:

- LSB-disentangled representation of dimension 4: Forward-VAE trained as in Sec.6.2.
- SB-disentangled representation of dimension 2: CCI-VAE variant trained as in Sec.6.1.
- Non-disentangled representation of dimension 2: Auto-encoder, non-disentangled baseline.
- SB-disentangled representation of dimension 4: CCI-VAE trained as in Sec.6.1 but with 4 dimensions, baseline to control for the effect of the size of the representation.

For each model, once trained, we created a dataset of transitions in the corresponding representation space $(s_t, a_t, s_{t+1})$. We then report the 10-fold cross-validation mean accuracy as a function of the maximum depth parameter of random forest, which controls the capacity of the classifier.

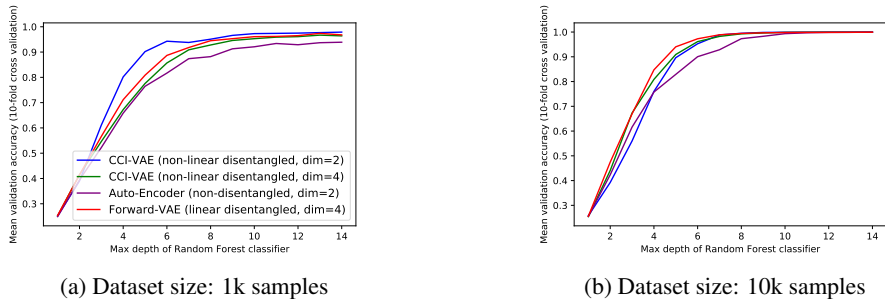

(a) Dataset size: 1k samples          (b) Dataset size: 10k samples

Figure 3: Downstream task evaluation of representation models: inverse model prediction. Mean 10-fold cross validation accuracy as functions of dataset size and classifier capacity (max depth parameter of Random Forest). LSB and SB-disentangled representation perform best.

## 7.2  Results

We first observe that in all cases, either LSB or SB-disentangled representations are performing best. In terms of final performance, all models meet at the upper 100% accuracy limit, given enough data and a classifier with enough capacity.

However, if we consider a constraint in training set size and a fixed high capacity classifier (see Fig.3), we can see that using a SB-disentangled representation is superior to other options. We refer to the capacity of the classifier as "high" if increasing the capacity parameter does not lead to an increase in validation accuracy.

Moreover, if we consider a fixed large training set size and a constraint on the classifier's capacity, using LSB-disentangled representation is the best option.

As a conclusion, we observed that it is easier for a small capacity classifier to solve the task using a LSB-disentangled representation and it is easier to solve the task using less data with a SB-disentangled representation. This indicates that (L)SB-disentanglement is indeed useful for downstream task solving.

### 7.3 Remarks

It's worth noting that the advantage is not very substantial, which is expected due to the simplicity of the task. Our results on usefulness of (L)SB-disentangled representations for downstream tasks are preliminary, it would be interesting as future work to compare to more baselines and on more tasks. Other related works such as van Steenkiste et al. (2019); Locatello et al. (2019a) also study the usefulness of disentangled representations for downstream tasks, and respectively find them useful for performance in abstract visual reasoning tasks and for encouraging fairness when sensitive variables are not observed.

More generally, there is a lack of large-scale evaluations of representations' usefulness for downstream tasks in the disentanglement representation learning literature. Such studies are needed to validate the intuition that disentanglement is useful in practice for subsequent tasks.

## 8 Discussion & Conclusion

**Discussion.** The benefit of using transitions rather than still observations for representation learning in the context of an agent acting in a environment has been proposed, discussed and implemented in previous work (Thomas et al., 2017; Raffin et al., 2019). In this work however, we emphasize that using transitions is not only a beneficial option, but is compulsory in the context of the current definition of SBDRL for an agent acting in an environment, as Theorem 1 proves it.

Applying SBDRL to more complex environments is not straightforward. For instance consider that we add an object in the environment studied in this paper. Then the group structure of the symmetries of the world are broken when the agent is close to the object. However, the symmetries are conserved locally. One approach would be to start from this local property to learn an approximate SB-disentangled representation.

**Conclusion.** Using theoretical and empirical arguments, we demonstrated that SBDRL (Higgins et al., 2018), a proposed definition for disentanglement in Representation Learning, requires interaction with the environment. We then proposed two methods to perform SBDRL in practice, both of which are successful empirically. We believe SBDRL provides a new perspective on disentanglement which can be promising for Representation Learning in the context of an agent acting in a environment.

## Acknowledgements

We thank Irina Higgins for insightful mail discussions.

## Footnotes

[1]https://colab.research.google.com/drive/1KVlSV24c687N_4TLJWwGTkjt3sh9ufWW

[2]https://github.com/Caselles/NeurIPS19-SBDRL

[3]https://colab.research.google.com/drive/1KVlSV24c687N_4TLJWwGTkjt3sh9ufWW

[4]https://github.com/Caselles/NeurIPS19-SBDRL

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
