[Supplementary Material]

# A   Proofs

## A.1   Symmetry-Based Disentangled Representation Learning requires interaction with environments

We prove Theorem 1.

**Theorem.** *Suppose we have a SB representation $(f, \cdot_Z)$ of a world $\mathcal{W}_0 = (W = (w_1, .., w_m) \in \mathbb{R}^{m \times d}, \cdot_{\mathcal{W}_0})$ w.r.t to $G = G_1 \times ... \times G_n$ using a training set $\mathcal{T}$ of unordered observations of $\mathcal{W}_0$. Let $W_k$ be the set of possible values for the $k^{th}$ dimension of $w \in W$.*
*Then:*

1. *There exists at least $k_{W,G} = n[(\min_k(card(W_k))!] - 1$ worlds $(\mathcal{W}_1, .., \mathcal{W}_{k_{W,G}})$ equipped with the same world states $\mathcal{W}_i = (w_1, .., w_m)$ and symmetries G, but different group actions $\cdot_{\mathcal{W}_i}$.*

2. *For these worlds, $(f, \cdot_Z)$ is not a SB representation.*

3. *These worlds can produce exactly the same training set $\mathcal{T}$ of still images.*

*Proof.*   We prove the three points.

1. For each symmetry $G_i$, we can shuffle the order of states along each axis of $W$. For instance, if the symmetry is translation along a cyclic hue axis composed of three colors (red, green, blue). Then one can consider two worlds where translating right from red moves the agent in blue (world 1) or green (world 2).

We provide a lower bound to the number of possible worlds. For a symmetry $G_i$, the minimal number of possible visited states is $\min_k(card(W_k))$. It is the case if all symmetries affect only one axis of $W$ and all axis of $W$ have the same number of possible values $(= \min_k(card(W_k)))$. The number of possible world is then given by the number of permutations of a set composed of $\min_k(card(W_k))$ elements, which is $\min_k(card(W_k))!$.

There are $n$ symmetries in $G = G_1 \times .. \times G_n$, hence there are at least $k_{W,G} = n[(\min_k(card(W_k))!] - 1$ possible worlds $(\mathcal{W}_1, .., \mathcal{W}_{k_{W,G}})$ that are not $\mathcal{W}_0$ but share the same state space $W$ and symmetries $G$. They differ by the action $\cdot_{\mathcal{W}_i}$ of $G$ on the world $\mathcal{W}_i$.

2. For any different world $\mathcal{W}_i$ than $\mathcal{W}_0$, there exists a state and a symmetry $(g, w \in G \times W)$ such that the action of $g$ on $w$ is not the same on the two worlds. Thus, $f$ is not equivariant between the group actions on $W$ and $Z$ w.r.t to both $\mathcal{W}_0$ and $\mathcal{W}_i$. Hence $(f, \cdot_Z)$ is necessarily not a SB representation w.r.t to any of the worlds $(\mathcal{W}_1, .., \mathcal{W}_{k_{W,G}})$ and $G$.

Formally, let $i \in [|1..k_{W,G}|]$. $\mathcal{W}_i \neq \mathcal{W}_0 \implies \exists (g, w \in G \times W)$, $g \cdot_{\mathcal{W}_i} w \neq g \cdot_{\mathcal{W}_0} w$. Necessarily, $f(g \cdot_{\mathcal{W}_i} w) \neq f(g \cdot_{\mathcal{W}_0} w)$. Yet, $(f, \cdot_Z)$ is SB w.r.t $\mathcal{W}_0$: $f(g \cdot_{\mathcal{W}_0} w) = g \cdot_Z f(w)$. Hence, $f(g \cdot_{\mathcal{W}_i} w) \neq g \cdot_Z f(w)$, i.e. for world $\mathcal{W}_i$, $(f, \cdot_Z)$ is not equivariant between the group actions on $W$ and $Z$.

3. $(\mathcal{W}_0, .., \mathcal{W}_{k_{W,G}})$ all share the same state space. Hence they can theoretically produce any training set of still images collected in $\mathcal{W}_0$.

$\square$

## A.2   Trivial representations

We first define trivial representations and then prove that they are LSB-disentangled. We will then use this definition to prove Theorem 2.

**Definition 1.** *$Z$ is a trivial representation if and only if $f$ is constant.*

If $Z$ is a trivial representation, we thus have that each state of the world $w \in W$ has the same representation.

**Proposition 1.** *If $Z$ is a trivial representation then $Z$ is LSB-disentangled w.r.t to every group decomposition.*

We prove Proposition 1 which states that trivial representations are LSB-disentangled.

*Proof.* The definition of LSB-disentangled representation of dimension 2 is:

1. There is a linear action $\cdot_Z : G \times Z \to Z$. It thus can be viewed as a group representation $\rho : G \to GL(Z)$.

2. The map $f : W \to Z$ is equivariant between the actions on $W$ and $Z$.

3. There is a decomposition $Z = Z_1 \times Z_2$ or $Z = Z_1 \bigoplus Z_2$ such that each $Z_i$ is fixed by the action of all $G_j, j \neq i$ and affected only by $G_i$.

Let $\rho(g)$ be the identity function $\forall g \in G$, which is linear.

We have that $f : W \to Z$ is constant. We can verify that $f$ is equivariant between the actions on $W$ and $Z$:

$$\rho(g)(f(w)) = f(w) = f(g \cdot_W w) \tag{3}$$

Finally, $Z$ has the same representation $\forall w \in W$, so $Z$ is fixed by the action of any subgroup of $G$. Hence for all decomposition of $G$, point 3. of the definition is satisfied.

$\square$

## A.3   It is impossible to learn a LSB-disentangled representation of dimension 2 in the considered environment

We prove Theorem 2 which states that it is impossible to learn a LSB-disentangled representation of dimension 2 in the environment presented in Sec.4 (the result also applies to the environment considered in Higgins et al. (2018)).

**Theorem 2.** *For the considered world, there exists no LSB-disentangled representation $Z$ w.r.t to the group decomposition $G = G_x \times G_y$, such that $\dim(Z) = 2$ and $Z$ is not trivial.*

*Proof.* Proof by contradiction.
Suppose that there exists a LSB-disentangled representation $Z$ w.r.t to the group decomposition $G = G_x \times G_y$, such that $\dim(Z) = 2$. Then, by definition:

1. There is a linear action $\cdot_Z : G \times Z \to Z$. It thus can be viewed as a group representation $\rho : G \to GL(Z)$.

2. The map $f : W \to Z$ is equivariant between the actions on $W$ and $Z$.

3. There is a decomposition $Z = Z_1 \times Z_2$ or $Z = Z_1 \bigoplus Z_2$ such that each $Z_i$ is fixed by the action of all $G_j, j \neq i$ and affected only by $G_i$.

We now prove that if these conditions are verified, $f$ is necessarily constant. Consequently, $Z$ has the same representation for each state of the world, which is a trivial representation. So, if $Z$ a LSB-disentangled representation of dimension 2 w.r.t to $G$, then $Z$ is the trivial representation.

We thus suppose that there exists a LSB-disentangled representation $Z$ of dimension 2 w.r.t to the group decomposition $G = G_x \times G_y$. Hence, we have, by point 2. of the definition:

$$g \cdot_Z f(w) = f(g \cdot_W w) \tag{4}$$

Since $\cdot_Z$ is linear, we can view it as a group representation $\rho$, as mentioned in point 1. of the definition:

$$g \cdot_Z f(w) = \rho(g)(f(w)) \tag{5}$$

Because $f(W) \in Z \subset \mathbb{R}^2$ and $W = W_x \bigoplus W_y = (x, y)$, we can re-write $f$ as:

$$\begin{aligned} f(w) &= f((x,y)) \\ &= (f_1(x,y), f_2(x,y)) \end{aligned} \tag{6}$$

Hence, combining (4) and (5):

$$f(g \cdot_W (x,y)) = \rho(g)((f_1(x,y), f_2(x,y))) \tag{7}$$

We can decompose any $g \in G$ into the composition of functions of each subgroup of $G$, i.e. $\forall g \in G = G_x \times G_y, \exists (g_x, g_y) \in G_x \times G_y$ such that $g = g_x \circ g_y$. Plus, by definition of $Z$ and because $W = W_x \bigoplus W_y = (x,y)$, the action of all $G_i$ on $W$ and $Z$ is fixed by the action of all $G_j, j \neq i$ and affected only by $G_i$. We can thus re-write both terms of Equation (7).

$$f(g \cdot_W (x,y)) = (f_1((g_x(x), g_y(y))), f_2((g_x(x), g_y(y))) \quad \text{since } g \cdot_W (x,y) = (g_x(x), g_y(y)) \tag{8}$$

$$\rho(g)((f_1(x,y), f_2(x,y))) = (\rho_x(g_x)(f_1(x,y)), \rho_y(g_y)(f_2(x,y))) \quad \text{by definition of } \rho \tag{9}$$

Hence, Equation 7 becomes:

$$(f_1((g_x(x), g_y(y))), f_2((g_x(x), g_y(y))) = (\rho_x(g_x)(f_1(x,y)), \rho_y(g_y)(f_2(x,y))) \tag{10}$$

We will now prove that $f_1$ is necessarily constant. The same argument applies for $f_2$.

From Equation (10), we have:

$$f_1((g_x(x), g_y(y))) = \rho_x(g_x)(f_1(x,y)) \tag{11}$$

$g_x$ and $g_y$ are respectively translations on the $x$-axis and $y$-axis. Let $N$ be the size of the grid, then $\exists (n_x, n_y) \in [|0, N|]$ s.t. $(g_x(x), g_y(y)) = ((x + n_x) \mod N, (y + n_y) \mod N)$. When at edge of the world, if the object translates to the right, it returns to the left, hence the modulo operation that represents this cycle. Hence:

$$\begin{aligned} f_1((g_x(x), g_y(y))) &= f_1(((x + n_x) \mod N, (y + n_y) \mod N)) \\ &= \rho_x(g_x)(f_1(x,y)) \end{aligned} \tag{12}$$

The key argument of the proof lies in the fact that $\rho_x(g_x)$ is necessary cyclic of order $2N$ (the minimal order can be inferior to $N$, but it is not useful to caracterize the minimal order in this proof). Let's compose $\rho_x(g_x)$ $2N$ times:

$$\begin{aligned} \rho_x(g_x)^{(2N)}(f_1(x,y)) &= f_1(((x + 2N \cdot n_x) \mod N, (y + 2N \cdot n_y) \mod N)) \\ &= f_1((x,y)) \end{aligned} \tag{13}$$

We now use the fact that $\rho_x(g_x)$ is a linear application of $\mathbb{R}$, thus:

$$\rho_x(g_x) \in GL(\mathbb{R}) \implies \exists (a(g_x), b(g_x)) \in \mathbb{R}^2 \quad s.t. \quad \forall x \in \mathbb{R} \quad \rho_x(g_x)(x) = a(g_x) \cdot x + b(g_x) \tag{14}$$

For notation purposes, we drop the dependence on $g_x$ of the coefficients of the real linear application $\rho_x(g_x)$, and we can rewrite Equation (10):

$$\rho_x(g_x)(f_1(x,y)) = a \cdot f_1(x,y) + b \tag{15}$$

Hence, using Equation 13 we can develop the term $\rho_x(g_x)^{(2N)}(f_1(x,y))$:

$$\begin{aligned} \rho_x^{2N}(g_x)(f_1(x,y)) &= a^{2N} \cdot f_1(x,y) + b \cdot \sum_{i=0}^{2N-1} a^i \\ &= f_1(x,y) \end{aligned} \tag{16}$$

Define $c = b \cdot \sum_{i=0}^{2N-1} a^i$, we have:

$$a^{2N} \cdot f_1(x, y) + c = f_1(x, y)$$
$$\iff \quad (a^{2N} - 1) \cdot f_1(x, y) + c = 0 \tag{17}$$

Equation (17) is verified $\forall (x, y) \in \mathbb{R}^2$. Let $((x_1, y_1), (x_2, y_2)) \in \mathbb{R}^2 \times \mathbb{R}^2$:

$$\begin{cases} (a^{2N} - 1) \cdot f_1(x_1, y_1) + c = 0 \\ (a^{2N} - 1) \cdot f_1(x_2, y_2) + c = 0 \end{cases} \implies (a^{2N} - 1) \cdot (f_1(x_1, y_1) - f_1(x_2, y_2)) = 0 \tag{18}$$

We can now derive conditions on $f_1$ or $(a, b)$. From Equation (18) we know that either $f_1$ is constant or $(a^{2N} - 1) = 0 \implies a = 1$. If $a = 1$, then Equation (17) simplifies to $c = 0 \implies b = 0$. So either $\rho_x(g_x)$ is the identity function or $f_1$ is constant. The same argument applies to $f_2$ and $\rho_y(g_y)$, hence we have that either $f$ is constant or $\rho(g) = \text{Id}(\mathbb{R}^2)$. By plugging the second option in Equation (7), we have that $\rho(g) = \text{Id} \implies f$ is constant.

Hence $f$ is necessarily constant, which implies that $Z$ is a trivial representation.

$\square$

# B  Hyperparameters and neural networks architectures

The code for our experiments is available at the following link: `https://github.com/Caselles/NeurIPS19-SBDRL`.

More specifically, the architecture and hyperparameters used for all the VAEs is available here: `https://github.com/Caselles/NeurIPS19-SBDRL/blob/master/code/learn_4_dim_linear_disentangled_representation/vae/arch_torch_sans_cos_sin.py`.

All representation are learned using the same base architecture mentioned above. For the Forward-VAE model, we only add the action matrices mentioned in the description of the model.

As for the training hyperparameters:

- We use 15k transitions for training, batch sizes of 128, $\beta$ annealed from 1 using a factor of 0.995 at each batch.

- For optimization, we use Adam (Kingma and Ba, 2014) with the standard hyperparameters provided in PyTorch (Paszke et al., 2017).

- LSB-disentangled representation of dimension 4 (Forward-VAE trained as in Sec.6.2): 35 epochs.

- SB-disentangled representation of dimension 2 (CCI-VAE variant trained as in Sec.6.1): 11 epochs.

- Non-disentangled representation of dimension 2 (Auto-encoder, non-disentangled baseline): 11 epochs.

- SB-disentangled representation of dimension 4 (CCI-VAE trained as in Sec.6.1 but with 4 dimensions, baseline to control for the effect of the size of the representation): 11 epochs.

As for the experiments in Sec.7, we use the standard implementation of random forest in Scikit-Learn, and we only modify the hyperparameters indicated in the experiments.

# C Details about Forward-VAE

## C.1 Definition of $\hat{A}$

$A^*(g)$ is a 2x2 block-diagonal rotation matrix of dimension 4. For instance, if $g = g_x \in G_x$ is a translation on the x-axis, the corresponding matrix is: $A^*(g_x) = \begin{bmatrix} \cos(n_x) & -\sin(n_x) & 0 & 0 \\ \sin(n_x) & \cos(n_x) & 0 & 0 \\ 0 & 0 & 1 & 0 \\ 0 & 0 & 0 & 1 \end{bmatrix}$.

Similarly, for $g = g_y \in G_y$ which is a translation on the y-axis, the corresponding matrix is $A^*(g_y) = \begin{bmatrix} 1 & 0 & 0 & 0 \\ 0 & 1 & 0 & 0 \\ 0 & 0 & \cos(n_x) & -\sin(n_x) \\ 0 & 0 & \sin(n_x) & \cos(n_x) \end{bmatrix}$.

Let's consider the environment in Sec.4. The agent has 4 actions: go left, right, up or down. We associate each action with a corresponding matrix with trainable weights. Thus, we associate actions go up and go down with a matrix of the form $\hat{A} = \begin{bmatrix} \cdot & \cdot & 0 & 0 \\ \cdot & \cdot & 0 & 0 \\ 0 & 0 & 1 & 0 \\ 0 & 0 & 0 & 1 \end{bmatrix}$, and we associate actions go left and go right with a matrix of the form $\hat{A} = \begin{bmatrix} 1 & 0 & 0 & 0 \\ 0 & 1 & 0 & 0 \\ 0 & 0 & \cdot & \cdot \\ 0 & 0 & \cdot & \cdot \end{bmatrix}$ where $\cdot$ represents trainable parameters.

## C.2 Pseudo-code of Forward-VAE

---
**Algorithm 1** Pseudo-code for training procedure of Forward-VAE

---
1: batch $= ((o_t, .., o_{t+k}), (a_t, .., a_{t+k-1})) = (\mathbf{o}, \mathbf{a})$
2: **for** batch in dataset **do**
3:
&emsp;&emsp;&emsp;&emsp;&emsp;&emsp;&emsp;&emsp;&emsp; *— Forward model Loss—*

4:
5: &emsp;&emsp; $\mathbf{z} \leftarrow encoder\_mean(batch)$
6: &emsp;&emsp; $\mathbf{z_{before}} \leftarrow z[: -1]$
7: &emsp;&emsp; $\mathbf{z_{after}} \leftarrow z[1 :]$ *# targets*
8: &emsp;&emsp; $\hat{\mathbf{A}} \leftarrow [\hat{A}(a_t), .., \hat{A}(a_{t+k-1})]$ *# actions matrices corresponding to given action sequence*
9: &emsp;&emsp; $\mathbf{z_{prediction}} \leftarrow \hat{\mathbf{A}} \cdot \mathbf{z_{before}}$ *# predictions*
10: &emsp;&emsp; $\mathcal{L}_{forward}(batch) \leftarrow MeanSquaredError(\mathbf{z_{prediction}}, \mathbf{z_{after}})$
11:
&emsp;&emsp;&emsp;&emsp;&emsp;&emsp;&emsp;&emsp; *— VAE Loss (reconstruction and KL) —*

12:
13: &emsp;&emsp; $\mathbf{z} \leftarrow encoder\_sample(batch)$
14: &emsp;&emsp; $\mathbf{\hat{o}} \leftarrow decoder(\mathbf{z})$
15: &emsp;&emsp; $\mathcal{L}_{recon}(batch) \leftarrow MeanSquaredError(\mathbf{\hat{o}}, \mathbf{o})$
16: &emsp;&emsp; $\mathcal{L}_{KL}(batch) \leftarrow KL\_divergence(\mathbf{z}, \mathcal{N}(0, 1))$
17:
&emsp;&emsp;&emsp;&emsp;&emsp;&emsp;&emsp;&emsp;&emsp; *— Backpropagation —*

18:
19: &emsp;&emsp; $\mathcal{L}_{Forward-VAE}(batch) \leftarrow \mathcal{L}_{recon}(batch) + \mathcal{L}_{KL}(batch) + \mathcal{L}_{forward}(batch)$
20: &emsp;&emsp; $encoder, decoder, (\hat{A}_1, .., \hat{A}_j) \leftarrow Backpropagation(\mathcal{L}_{Forward-VAE}(batch))$

---

# D  Additional results

We observe that the mean squared difference between the ideal matrices $(A_i)_{i=1..4}$ and the learned matrices $(\hat{A}_i)_{i=1..4}$ is very small (order of $10^{-4}$). Hence, we have :

$$\hat{A}(\text{go left / go right}) \approx A^*(\text{go left / go right}) = \begin{bmatrix} \cos(\pm\alpha) & -\sin(\pm\alpha) & 0 & 0 \\ \sin(\pm\alpha) & \cos(\pm\alpha) & 0 & 0 \\ 0 & 0 & 1 & 0 \\ 0 & 0 & 0 & 1 \end{bmatrix}$$

$$\hat{A}(\text{go up / go down}) \approx A^*(\text{go up / go down}) = \begin{bmatrix} 1 & 0 & 0 & 0 \\ 0 & 1 & 0 & 0 \\ 0 & 0 & \cos(\pm\alpha) & -\sin(\pm\alpha) \\ 0 & 0 & \sin(\pm\alpha) & \cos(\pm\alpha) \end{bmatrix}$$

The result is quite surprising as we do not have completely explicitly optimized for this matrix (at least for the cos/sin part). Plus there is no instability in training.

One issue with the fact that the approximation is not exact, is unstability with composition. Rotation matrices' determinants are stable with composition, as we have:

$$\det(AB) = \det(A)\det(B)$$

As rotation matrices have a determinant equal to $1$, the composition operation is cyclic for rotations.

However, as $(\hat{A})_{i=1..4}$ are only approximation of rotation matrices, their determinant is approximately $1$ but not exactly. This is why, as many compositions are performed, the determinant of the resulting matrix either collapses to zero or explodes to $+\infty$. We provide evidence for this phenomenon in Fig.4.

Figure 4: Determinant of real ($A^*$) and learned rotation matrix ($\hat{A}$) as a function of number of compositions. As many compositions are performed, the determinant of the approximation of the rotation matrix $A^*$ collapses to zero.