[Reviews · NeurIPS 2019]

Reviewer 1



* Update after reading Author Response * Thanks to the authors for addressing my concerns. After reading the other reviews and the author feedback, I believe the changes the authors propose to make will strengthen the paper. I believe the ideas in this paper have value and it is a positive step to establish some theoretical results in this space. I am still concerned with the clarity of the paper, especially for the broader community that might not be very familiar with the disentangled representation literature. With that in mind, I revise my score upward to 6. * Original * Overall, while the paper attempts to tackle some interesting theoretical questions, it is held back by unclear proofs, limited and inconclusive experiments, and lack of novel insight. The work here on its own does not meet the standard for publication. Below are some more specific comments. There seems to be a fundamental issue here, which is the notion that a symmetry is an action that the agent can take, rather than a geometric property of the state space. It would be helpful to clarify exactly what notion of symmetry the group actions are supposed to capture. The paragraph starting at line 112 is confusing. First, the examples of what the group action might be are a bit vague. Second, in line 117 it seems you require the group action to be the dynamics function? Why should the group action on the observation at time t be equal to the observation at time t+1? A symmetry encodes a geometric constraint or property about the state space; it shouldn't be interpreted as an action applied by an agent. The statement of Theorem 1 is confusing. From your notation, it appears that world states w_i are themselves sets? How are they to be interpreted? What does it mean "using a training set T of still images"? The detailed proof in the appendix is likewise confusing. There appear to be mismatches in notation (cf line 17 in the appendix and line 122 in the main text -- are we considering the cardinality of a world W_i or a world state w_i?). The reference to shuffling the order of states along an axis also doesn't make sense -- there seems to be some major missing details here. The entire proof is difficult to evaluate since it's not at all clear what is being argued. Theorem 2: what is the significance of this result? The experimental results seem inconclusive about which approach is better, since, as the authors note, the task is so simple.

Reviewer 2



This paper provides theoretical results showing that in order to learn disentangled representations grounded in symmetry transformations, as recently defined by Higgins et al (2018), it is necessary to have access to the actions producing observation changes. It them proposes a new model architecture that is able to exploit this information to learn symmetry based disentangled representations (SBDR). Finally, the paper demonstrates that disentangled representations improve sample efficiency for inverse model learning. The work is a first step building on the recent theoretical paper by Higgins et al (2018) that defines disentangling in terms of symmetry transformations. Hence, this work is an important step that other can build on. The submission is technically sound and provides both theoretical and empirical contributions. However, the empirical contributions are quite limited, since the authors only use one very simple dataset. I would have liked to see the approach evaluated on more challenging datasets too. Saying this, the paper is very well written and I think it makes a significant enough contribution to the field to be published in providing a proof of principle demonstration of how to address a challenging problem (learning SBDR). ---- Post author feedback ------ Thank you for your detailed feedback. I leave my score unchanged given that it was favourable to start with.

Reviewer 3



The authors adopt the recently proposed symmetry-based disentangled representation learning (SBDRL) framework by Higgins et al. Although this particular view of disentangled representations has not yet been peer-reviewed, the theory appears sound and offers an interesting perspective. SBDRL is focused on transformation properties of the world, and defines disentangled representations according to two criteria. (1) The map f, which maps world-states to representations is equivariant between group actions on the world states and group actions on the representations. (2) There is a decomposition of representational space into subspaces that remain unaffected by all group actions, but a specific group action that is associated with that subspace. The theoretical contributions of this paper primarily focus on this first criteria. The authors notice that learning from static images (i.e. corresponding to rendered world states) does not provide a means to recover the effect of a group-action on a particular state of the world. Since different worlds having the same states and symmetries that vary only in the effect of their group actions on the states (here different ways of cycling through a list is provided as an example) can give rise to the same observations, criteria (1) can not be verified for a learned mapping from static images. Although the proof is not particularly complex, it is both insightful and thought-provoking. The authors conclude based on this insight that one must have access to state transitions of the form (o_t, a_t, o_t+1) to learn a mapping that satisfies criteria (1). Another interpretation, which should be discussed, is that the SBDRL framework is too strict in the requirements that it imposes on the learned mapping. While it is intuitive that interactions are important in learning about symmetries in the environment, it is not clear that the learned mapping has to be able to differentiate between different worlds that have the same states and symmetries. In the remainder the authors provide a simple modification to a VAE that allows it to learn LSB-disentangled representations from interactions data. The environment considered is similar to the one in Higgins et al. except that it contains one less dimension (the color). Theorem 2 proves that a non-trivial LSB-disentangled representation of dimensionality 2 can not be learned for the symmetries (x, y) in this environment. Personally I do not find this result particularly significant, as it is intuitive that this can not be done. I would therefore recommend moving section 5 to the appendix and dedicate this space to something else. In the first experiment symmetry-based disentangled representations are learned on this environment. First a variant of CCI-VAE is trained to learn an SB-disentangled representation on the static frames, and afterwards a separate network that learns the group action from the interaction data. I would like to mention that the fact that this is possible (i.e. decoupling these two parts) suggests that differentiating between different worlds with the same symmetries may not be important in learning disentangled representations (in the context of other definitions). Details about this experiment are missing and should be added, these include hyper-parameters but also a clear description of the architecture used. The second experiment aims at learning an LSB-disentangled representation using an end-to-end approach that incorporates an additional loss to anchor the effect of the group action on the representations. A strong inductive bias (different block-diagonal matrices for different actions) is used to achieve this, and I would have liked to see a more general approach, in which the entire matrix is learned (rather than only a subset of the parameters). Again important experiment details are missing and should be added (hyper-parameters, architectures, etc.). In the remarks part (section 6.3) it is argued that Forward-VAE learns even in the presence of potential instability. There is no (empirical) evidence provided in the paper for this claim, which should be added. The final experiment evaluates the various learned representations on a down-stream task in which an inverse model needs to be recovered. The choice of task is logical, and I think that this is an interesting experiment. However hyper-parameters are again missing, which makes it difficult to assess the overall validity of the comparison. In general it is found that disentangled representations are better than entangled ones, and that SB-disentangled is better than LSB-disentangled in the few-sample regime, while the opposite is true in the many-sample regime. This last result is unintuitive to me and should be commented on. An additional experiment with a different down-stream classifier would also help in shedding further light on this. Note that the conclusion in lines 279-280 appears wrong, while 281-283 is correct. Overall I find this a strong paper of good quality that offers a significant contribution to the disentangled representation learning community. The clarity of the paper is good, although it should be improved by adding missing details and extra discussion as mentioned before. One additional concern that I have is the lack of mentioning prior work, and the references used. Importantly, arxiv references should not be used for peer-reviewed publications, and additional prior work should be incorporated on disentangled representation learning including the work by Eastwood et al. and Ridgeway et al. that offer other notions of disentanglement, related work on interactive perception (eg. Bohg et al.) and causality. Other notions of disentanglement should be discussed and put in perspective based on the findings presented in this paper. Finally, with disentangled representation learning being an active field, I would encourage the authors to incorporate prior work that has come out since the submission deadline. In particular recent studies evaluating the benefits of disentangled representations (eg.van Steenkiste et al. (2019), Locattelo et al. (2019)) appear relevant. ####################### POST REBUTTAL ########################### I have read the author response and the other reviews. While the authors sufficiently address my initial critique (regarding literature review, and missing experimental details), I decided to leave my score unchanged as I find myself in agreement with the other reviewers regarding the clarity of the technical results presented. I encourage the authors to make all revisions required and promised as part of future updates.

[Author Response · NeurIPS 2019]

We thank all three reviewers for their helpful comments, which we attempt to answer hereafter.

**Theory.** About Reviewer #1's first comment on actions and symmetries, we are interested in practical ways to learn SBD representations, and make an analogy/parallel between the effect of a symmetry $g$ (by the group action $\cdot_{\mathcal{W}}$) on the environment $(o_1, g, g \cdot_{\mathcal{W}} o_1 = o_2)$, and a transition that uses the dynamics $f$ of the environment $(o_t, a_t, f(o_t, a_t) = o_{t+1})$. It allows us to consider an embodied scenario, where symmetries are applied via group actions to an agent evolving in an environment. In our analogy we simply state that $o_1 = o_t$, $o_2 = o_{t+1}$ and $a_t = g$ and $\cdot_{\mathcal{W}} = f$. However, we do not consider that symmetries and actions are always the same. A symmetry is an element of a group (in the mathematical sense) of functions $g : W \to W$, and the binary operation of the group is composition. In that sense, these functions can effectively be considered as actions, because actions take the environment from one state to another through the dynamics $f$, and symmetries take the environment from one state to another through the group action $\cdot_{\mathcal{W}}$. However it is important not to say that all actions are symmetries, for instance the action of eating a collectible item in the environment is not part of any group of symmetries of the environment because it might be irreversible for instance. This point is important to clarify in the paper, and we updated the paper in the beginning of Sec.3 with this explanation.

We thank Reviewer #1 for noticing the erroneous notation in claim 1 of Theorem 1. We clarify the notation: $\mathcal{W} = (W, \cdot_{\mathcal{W}})$ is a world, which comprises a set of states $W = (w_1, .., w_m)$, where each state $w_i$ is a d-dimensional vector, and a group action $\cdot_{\mathcal{W}}$ w.r.t a group $G$. In our example, $w_i$ is the position of the agent $(x, y) \in \mathbb{R}^2$. **In claim 1 of Theorem 1 we consider $W_k$ to be the set of possible values for the $k^{th}$ dimension of states** $w \in W$, e.g. all the possible values of $x$ would be $W_1$ for the aforementioned example. We hope this clarifies Theorem 1, which in claim 1 establishes a lower bound, as a function of the cardinalities of $(W_1, .., W_m)$, of the number of possibilities of how the group action $\cdot_{\mathcal{W}}$ can be applied to $W$. All these possibilities form a set of possible worlds $(\mathcal{W}_1, .., \mathcal{W}_{k_{W,G}})$ that have the same $W$ but different group actions. We added the definition of $W_k$ in the formulation of Theorem 1, and fixed claim 1. We also added a notation clarification in Sec.2, in order to introduce the useful notations earlier in the paper.

"Using a training set T of still images" refers to using only unordered observations for training, i.e. for learning a representation model. We clarified this in Theorem 1.

As suggested by Reviewers #1 and #3, Theorem 2 has been moved to the appendix. In the main text we still keep Sec.5 and mention the result in order to motivate the experiments that follows. The additional freed space has been used by the clarifications and additional experiments of the rebuttal.

**Experiments.** Following Reviewer's #3 suggestion, we added hyperparameter and architecture details for our experiments. Note that we did not include them at first because our experiments did not depend on hyperparameter and architecture search, as we used standard choices. Our code is also provided in the Google colab.

Regarding a more general approach for the Forward-VAE architecture (Reviewer #3), we indeed explicitly design the model such that the resulting representation is Linear SB-disentangled, because we enforce linearity, force the representation to be SB (see points 1 and 2 in the definition in Sec.3 and by design have two separate subspaces for each symmetry. A more general approach would have been not to have those two separated subspaces and learn the entire action matrices, and thus we won't have the guarantee that the representation will satisfy the disentangled property. We ran this additional experiment and obtained the expected result: the learned representation is Linear-SB but not disentangled. This means that the x and y coordinates are not properly disentangled w.r.t to the considered group decomposition (i.e. a latent traversal over each dimension would not result in only a movement of the agent along the x or y coordinate). Still, the learned action matrices are able to describe how the symmetries affect the representation and in a linear way. Enforcing disentanglement is the only viable option we found for LSB-disentanglement with this architecture. We added this additional experiment and conclusion in the remarks section (Sec.6.3) about this experiment.

The instability (line 244) mentioned by Reviewer #3 can be expected during training due to the different contributions of the loss: at each training steps the goal of the forward part of the loss is to have a latent space that is suited for predicting $z_{t+1}$ using $z_t$. The rest of the loss is the VAE, which tries to learn a latent space that allows reconstruction. Hence we considered the balance between these two seemingly unrelated objectives as a source of instability. However it worked in practice, without any reweighting of the objectives, which was a surprise. We rephrased the sentence.

**Related works.** It is indeed true that the paper is light on related work. Our initial intent was only to extend the work of Higgins et al., and so we omitted most of the related work in the paper, and pointed to the work of Higgins et al. for context. However, as suggested by Reviewer #3, better positioning the paper in the disentangled representation learning field might prove helpful for readers. The works mentioned by Reviewer #3, which we are aware of and agree are relevant in our paper, are now included in a paragraph in the Discussion section. This way, it is easier to map our paper in the current state of the art of the active field of disentangled representation learning. Finally, we updated all references that pointed to the arxiv version of the paper rather than the peer-reviewed one.

[Meta-Review · NeurIPS 2019]

I would recommend this paper for acceptance. The paper considers the (relatively new) idea of finding symmetry-based disentangled representations and provides a fresh perspective on the settings where such representations may be learned. As such, this paper stands apart from the numerous paper investigated in the last year that propose marginal improvements to existing disentanglement metrics and methods. While reviewers have brought up that the exposition can be improved, the authors have responded to these concerns in the author response and are encouraged to further polish their submission for the camera-ready version.